# Numerical Study and Optimal Design of the Butterfly Coil EMAT for Signal Amplitude Enhancement

**DOI:** 10.3390/s22134985

**Published:** 2022-07-01

**Authors:** Jingjun Zhang, Min Liu, Xiaojuan Jia, Ruizhen Gao

**Affiliations:** College of Mechanical and Equipment Engineering, Hebei University of Engineering, Handan 056038, China; zhangjingjun@hebeu.edu.cn (J.Z.); liumin980126@163.com (M.L.); gaoruizhen@hebeu.edu.cn (R.G.)

**Keywords:** EMAT, signal amplitude, butterfly coils, energy conversion efficiency, orthogonal test theory

## Abstract

The low energy conversion efficiency of electromagnetic acoustic transducers (EMATs) is a critical issue in nondestructive testing applications. To overcome this shortcoming, a butterfly coil EMAT was developed and optimized by numerical simulation based on a 2−D finite element model. First, the effect of the structural parameters of the butterfly coil EMAT was investigated by orthogonal test theory. Then, a modified butterfly coil EMAT was designed that consists of three−square permanent magnets with opposite polarity (TSPM−OP) to enhance the signal amplitude. Finally, the signal amplitude obtained from the three types of EMATs, that is, the traditional EMAT, the EMAT optimized by orthogonal test theory, and the modified EMAT with TSPM−OP, were analyzed and compared. The results show that the signal amplitude achieved by the modified butterfly coil EMAT with TSPM−OP can be increased by 4.97 times compared to the traditional butterfly coil EMAT.

## 1. Introduction

Non−destructive testing (NDT) is an inspection method that does not damage the physical structure and material properties of the test specimen [1]. It is widely used for industry and structural health monitoring, such as evaluating the quality of continuously cast high carbon billets, and inspection of railway tracks, tank floors, oil and gas pipelines, and large metal plates. NDT methods mainly include ultrasonic testing (UT), leak testing (LT), AC field measurement (ACFMT), penetration testing (PT), magnetic particle testing (MT), eddy current testing (ECT), radiographic testing (RT), acoustic emission testing (AET), and magnetic flux leakage (MFL) inspection. However, RT, ECT, and MFL examinations are flawed due to radiation exposure and structural limitations [2,3]. UT, as one of the NDT technologies, has the characteristics of rapid detection, wide detection range, high detection accuracy, and good environmental adaptability. It is a method used to determine whether the inside or the surface of a specimen has defects, using the propagation characteristics of the ultrasonic wave, namely, reflection, transmission, and refraction [4]. UT is the most important of these approaches, with more than half of all inspections performed with one or other form of ultrasonic probe [5].

Piezoelectric ultrasonic testing is one of the most widely used UT methods. Although piezoelectric ultrasonic technology has obvious advantages, such as good detection capability and a strong signal, it requires a coupling agent and treatment of the specimen surface for detection, thus limiting the application of this transducer in high−temperature environments, and affecting the inspection’s accuracy and efficiency [6]. In the case of rail breakage, the traditional NDT methods have solved many problems relating to the detection of defects in the tread and web of the rails. However, the existence of complex structures and beveled corroded surfaces at the rail base makes it difficult for the traditional NDT methods to accurately detect cracks at the bottom of the rail base [7]. Electromagnetic acoustic transducers (EMATs) do not require direct contact with the specimen being tested during the detection process, and they outperform piezoelectric transducers without the need for coupling agents and surface pretreatment [8]. Furthermore, EMATs have the capability of operating at high temperatures and high speeds, and in other harsh environments, in addition to having significant advantages in thickness measurement and detection of rail defects due to their good penetration ability [7,9].

The main problem in developing EMATs is the low signal−to−noise ratio (SNR), which is due to poor energy conversion efficiency and a significant lift−off (between the coil and the specimen surface in the EMAT) effect [10]. The disadvantage of EMATs compared to conventional piezoelectric ultrasonic sensors is their low energy conversion efficiency, which is only about one percent [6]. Furthermore, the efficiency strongly decreases with increasing lift−off. A limited range for magnetic coupling between the inspected component and the EMAT is about 1 mm or less [11,12]. Therefore, improving the energy conversion efficiency of EMATs is vital and a significant challenge in the field of NDT. Experts and scholars in China and abroad have conducted a series of studies about how to improve the energy conversion efficiency of EMATs, and certain results have been achieved.

Improving energy conversion efficiency by increasing magnetic flux density is an extremely effective method. In this regard, finite element methods (FEMs) have been used to analyze the influence of parameters, such as coil size, the number of coil turns, and the structure and position of the permanent magnet, on the performance of magnetostrictive transducers [13]. Pei et al. [14] proposed a modified permanent magnet configuration for the Rayleigh wave EMAT, which allows the horizontal magnetic field intensity strength to be increased for the purpose of generating Rayleigh wave signals with larger amplitudes. Liu et al. [15] weakened the horizontal component of the magnetic flux density and the vertical Lorentz force. The A0 mode wave was weakened and the mode purity of the transducer was improved by adding a magnetic circuit aggregation device below the permanent magnet. Zhang et al. [16] designed a flux−concentrated EMAT with radial flux−focused permanent magnets to significantly enhance the static magnetic field strength. Sun et al. [17] designed a novel transducer structure, which is composed of a fan−shaped periodic permanent magnet (PPM) and a centripetal conductor. Compared with the conventional EMAT, the improved transducer is better able to focus the guided waves and enhance the signal. A novel permanent magnet arrangement structure that effectively increases the vertical flux density and peak echo signal without changing the permanent magnet thickness was proposed [18]. In order to excite and receive single−mode A0 waves with a high SNR, a new EMAT consisting of a coil with a contra−flexure structure and a permanent magnet was proposed [19]. Kang et al. [20] investigated the effect of different geometrical parameters of the surface wave EMAT on the energy conversion efficiency using orthogonal test theory, and the results showed that the signal amplitude could be improved by 25.2% for the EMAT after orthogonal optimization. Dutton [21] designed a new EMAT for enhancing the magnetic density applied to the tested specimen. The results show that the detection of in−plane displacement can be improved using the new EMAT. To increase the eddy currents and dynamic magnetic field strength in the specimen, a back−plate made of laminated silicon steel is placed between the coil and the permanent magnet [22].

It is equally feasible to improve the energy conversion efficiency by optimizing the EMAT geometric parameters. Previously, the optimal design of transducer geometrical parameters was performed using a univariate design method, i.e., changing a design variable and comparing the change in received signal strength. Fewer methods are available for exploring and comparing the influence levels of design factors for multiple factors and design objectives. The optimal design of EMATs involves a large number of computational steps and a huge number of calculations due to the parameter values of each component and excitation signal. Therefore, in the optimization design of EMATs, the optimization design method with the highest optimization−seeking efficiency should be selected as much as possible. The orthogonal test theory is a multi−factor and multi−level optimization design method based on mathematical and statistical theory. On the basis of orthogonality, this design method is tested by selecting some representative points from a comprehensive test. It is noteworthy that these representative points have uniform dispersion and neatly comparable characteristics. The orthogonal table is used as the main tool of orthogonal test theory, which is selected based on the number and level of factors in the test, and the representative points are selected from the comprehensive test, which can achieve results equivalent to those of a large number of full−scale tests using a minimum number of tests [23]. Therefore, orthogonal test theory is widely used. Lan et al. [24] investigated the effects of structural and electrical parameters of surface wave EMATs on the energy conversion efficiency using an orthogonal test method. The results show that the signal amplitude of EMATs is 3.48 times greater in−plane and 3.49 times greater out−of−plane compared with the original EMAT. Sun et al. [25] used orthogonal test theory for parameter optimization when considering the focusing intensity and focusing area to obtain the optimal combination of parameters. Jia et al. [26] investigated the degree of influence of several parameters of the transducer on the signal amplitude, axial focus offset, and radial focus offset using an orthogonal test method. Thus, the focusing performance of the PFSV−EMAT was improved.

The butterfly coil consists of an interior and an exterior, with the interior having a high field density and the exterior having a low field density. The unique feature of this design is that most of the magnetic energy and inductance is concentrated around the parallel, linear wires in the center. The current density on both sides of this coil is significantly lower (about one−fifth), and, thus, the magnetic energy density is much lower (about 1/25). As a result, the inductance of the wires concentrated in the center should be five times the inductance of the sides [11]. The butterfly coil has the highest eddy current and pressure at the central axis. By comparing the circular area of the spiral coil with this “hot area” of the butterfly coil, it can be determined that the butterfly coil produces greater coverage and more concentrated energy [27], assuming similar area and energy. This is of practical importance. Moreover, for the traditional EMAT designs (with a stationary field/permanent magnet), it has been shown that the use of butterfly coils can produce stronger eddy currents and higher Lorentz forces in the test specimen [28]. Parra−Raad et al. [29] presented an EMAT that contains two orthogonal co−located butterfly coils. The EMAT can excite two orthogonal polarized shear waves in a metallic material for detecting the presence of crack−like defects in metallic materials. Nevertheless, this optimization method only considers the magnetic flux applied to the center of the butterfly coil. In practice, currents are also present on the left and right sides of the butterfly coil. Therefore, it is of practical importance to consider the overall structure of the butterfly coil when conducting a study of the butterfly coil EMAT. Ashigwuike et al. [30] investigated and compared the induced currents and Lorentz force densities generated within the skin depth by different coil structures on four tubes considering the overall structure of the butterfly coil, and the results showed that, for long−range ultrasonic detection, the butterfly coil structure proved to be a better alternative. However, the effect of EMAT geometrical parameters on eddy currents and Lorentz forces was not addressed in the study. Zhang et al. [31] designed a bulk wave PE−EMAT, including a core, a butterfly coil, and a driving circuit for a pulsed solenoid. The experimental results show that the designed new EMAT has better transducer efficiency than the permanent magnet. However, it still usually fluctuates in the low energy conversion efficiency range compared to piezoelectric transducers. It is known that the SNR of an EMAT with a Lorentzian mechanism is proportional to the square of the static magnetic field [14]. However, the SNR is related to energy conversion efficiency. Therefore, increasing the static magnetic field is particularly beneficial.

In this work, a modified EMAT with TSPM−OP is proposed, in which the overall structure of a butterfly coil is considered and the influence of the geometric parameters of the EMAT on the energy conversion efficiency is investigated. The organization of this work is as follows. Section 2 describes the working principle of EMATs based on the Lorentz force mechanism. In Section 3, the development of the 2−D FEM of the butterfly coil EMAT is presented and the simulation results are analyzed. The optimal combination of parameters of the butterfly coil EMAT is obtained by orthogonal test theory in Section 4. Based on orthogonal test theory, a modified butterfly coil EMAT consisting of three−square permanent magnets with opposite polarity (TSPM−OP) is proposed, which results in the improvement of the static magnetic field, the Lorentz force, and the amplitude of the ultrasonic signal. This is presented in Section 5, and Section 6 provides the conclusion.

## 2. Working Principle of EMATs

According to different properties of the specimen, the following mechanisms are responsible for the electro−acoustic transduction of EMATs: the Lorentz force mechanism, the magnetostrictive force mechanism, and the magnetizing force mechanism [32]. For non−ferromagnetic materials such as copper and aluminum, the mechanism of ultrasonic waves is the Lorentz force. For ferromagnetic materials such as iron, cobalt, nickel, and steel, the three kinds of working mechanisms described above are all included. In this work, aluminum is selected as the tested specimen, so only the Lorentz force mechanism is considered.

The energy conversion mechanism of the butterfly coil EMAT is shown in Figure 1. When the butterfly coils are supplied with high−frequency and high−power excitation current $J_{f}$
_,_ it can be known from Ampere’s law that it will generate an alternating magnetic field around the aluminum plate. At the same time, eddy currents $J_{e}$ with the same frequency and different directions will be generated at the skin depth of the surface of the specimen. Under the action of the static magnetic field perpendicular to the surface of the aluminum plate provided by the permanent magnet, the eddy currents generate an alternating Lorentz force F, so that the particles under the surface of the specimen produce high−frequency vibration, which creates ultrasonic waves inside the aluminum plate. The receiving process is the inverse process of the excitation procedure.

In the butterfly coil EMAT, the static magnetic induction intensity B provided by the permanent magnet can be obtained from the following equation [33]:(1)$$\mu \nabla^{2}\phi_{m}=0$$
(2)$$H=-\nabla \phi_{m}$$
(3)$$B=\mu H+B_{r}$$
(4)$$\sigma_{m}=\mu M_{0}\cdot n$$
(5)$$M_{0}=B_{r}/\mu_{0}$$
where $\varphi_{m}$, H, and μ are the scalar magnetic potential, magnetic field strength, and permeability, respectively. $\sigma_{m}$ is the surface magnetic density of the permanent magnet, and $B_{r}$ is the residual magnetic induction density of the permanent magnet.

In the excitation butterfly coil EMAT, the control differential equation of the *k*th current−carrying wire of the butterfly coil is as follows:(6)$$\mathrm{div}\frac{1}{\mu }gradA_{z}-\sigma \frac{\partial A_{z}}{\partial t}+\frac{\sigma }{S_{k}}\frac{\partial }{\partial t}\iint_{R_{k}}A_{z}ds=-\frac{i_{k}(t)}{S_{k}}$$
where $A_{Z}$, $i_{k}(t)$, $R_{k}$, and $S_{k}$ denote MVP, total current, cross−sectional region of the *k*th conductor, and the cross−sectional area of the *k*th conductor, respectively; σ is conductivity.

In source−free conducting regions the MVP must satisfy [34]:(7)$$\mathrm{div}\frac{1}{\mu }gradA_{Z}-\sigma \frac{\partial A_{Z}}{\partial t}=0$$

The propagation equation of the bulk waves excited by the Lorentz force in the aluminum plate can be expressed as [35]:(8)$$\rho \frac{\partial^{2}u}{\partial t^{2}}-\nabla •T=f_{L}$$
where ρ, u and T are the mass density, the elastic deformation, and the elastic stress tensor respectively.

## 3. Finite Element Modeling and Simulation Analysis of EMATs

A 2−D FEM of the butterfly coil EMAT was established by COMSOL Multiphysics 5.2. As shown in Figure 2, the model consists of a permanent magnet, a butterfly coil, an aluminum plate, and an air domain. So as to reduce the influence of reflection on the signal amplitude, the left and right sides of the aluminum plate and its bottom side are set as low−reflection boundaries. The specific modeling parameters are shown in Table 1.

Due to the skin effect, the eddy current generated by the butterfly coil EMAT is primarily concentrated at the skin depth. Therefore, considering the accuracy and the amount of calculation of the 2−D FEM of the butterfly coil EMAT, it is necessary to refine the grid of coils, permanent magnets, and aluminum plates three times in the skin depth directly below the coils. The other areas are divided by a freely divided triangular mesh. The schematic of the 2−D model in COMSOL is shown in Figure 3.

The static magnetic field produced by the permanent magnet was calculated in the steady simulation, and the distribution of the static magnetic field was obtained and is shown in Figure 4. It is shown in Figure 4 that the static magnetic field distribution of the permanent magnet is stronger at the edge and weaker at the center, and the vertical magnetic field can be obtained below the center of the permanent magnet. The same model can also be used to calculate the signal amplitude of the butterfly coil EMAT in the X− and Y−components. Due to the fact that the particle vibrations in this model are mainly along the X−component, the X−component of particle vibration of the point P (7, −20) is chosen as the observation point. Figure 5 shows the simulated ultrasonic waveforms at point P on the model surface generated by the EMAT. The maximum signal amplitude of the X−component at point P is $5.288\times 10^{-8}$ mm.

A comparison of the simulation results of the EMAT with different parameters reveals that the signal intensity is influenced significantly by the structural parameters of the EMAT [20]. Therefore, the appropriate choice of the structural parameters of the butterfly coil EMAT is important for improving the energy conversion efficiency of EMATs.

## 4. Optimization of EMATs

### 4.1. Orthogonal Test Design

In order to obtain the optimal combination of transducer parameters for the energy conversion efficiency of the butterfly coil EMAT, the essential parameters affecting the energy conversion efficiency were extracted based on the established 2−D FEM, and the influence of each parameter on the energy conversion efficiency was analyzed as the basis of optimization. Due to a large number of computations and the large variety of EMAT parameter combinations, the orthogonal test theory was used. This is a design method used for multi-factor and multi-level studies, which can not only efficiently find the optimal combination of parameters for the butterfly coil EMAT, but also analyze the influence law of each parameter on the energy conversion efficiency. Therefore, the orthogonal test theory was selected to analyze the signal amplitude for the butterfly coil EMAT with different structures.

In this work, the signal amplitude was used as the test result of the orthogonal test design, and the seven EAMT’s parameters were taken into consideration, namely, the permanent magnet width $w_{2}$, the permanent magnet height $h_{2}$_,_ the coil width $w_{1}$, the coil height $h_{1}$, the coil number n, the coil inner diameter r, and the lift−off distance $d_{1}$.

The ranges of variations in the butterfly coil EMAT parameters, which are the factors of the array, were chosen according to a set of common and realistic specifications, and the local environment of the butterfly coil EMAT; these were $w_{2}$: 25~30 mm, $h_{2}$: 10~20 mm, $w_{1}$: 0.2~0.5 mm, $h_{1}$: 0.15~0.45 mm, n: 5~9, r: 1~2 mm, $d_{1}$: 0.1~0.3 mm. In Table 2, three levels are taken for each of these seven factors, and an orthogonal test design was carried out, using the orthogonal array L_18_ (3^7^) as shown in Table 3. The FEM was modified according to the 18 groups of data in Table 3, and the signal amplitudes of the ultrasonic waves at point P (7, −20) for each of the 18 cases were obtained after calculation, as shown in the last column of Table 3.

### 4.2. Analysis the Results of the Orthogonal Test Design

According to Table 3, the influence of each parameter on the signal amplitude and their preferred values can be obtained by orthogonal analysis, as shown in Table 4, where $k_{i}$(i = 1, 2, 3) represent the corresponding signal amplitude at each level of the parameters, and R denotes the difference between the largest and the smallest values of $k_{i}$. From Table 4, it can be seen that the parameter with the larger R has the most significant influence on the signal amplitude. The order of effect of each factor is $h_{2}$ (2.308) > n (2.271) > $w_{2}$ (1.039) > $h_{1}$ (1.003) > $w_{1}$ (0.98) > $d_{1}$ (0.964) > r (0.593).

Figure 6 shows the average value and influence degree of each factor on the different results. It can be found that the factors that have the greatest effect on the signal intensity are the permanent magnet height $h_{2}$ and the coil number n, followed by the permanent magnet width $w_{2}$, the coil height $h_{1}$, the coil width $w_{1}$, and the lift−off distance $d_{1}$, whereas the influence of the coil inner diameter r is the smallest. It can be seen that a decrease in $w_{2}$ and $d_{1}$, and an increase in $h_{2}$ and r, have a positive effect on the signal amplitude. In addition, when $w_{1}$ = 0.2, $h_{1}$ = 0.15, and n = 5, the signal amplitude is the largest. Therefore, in order to improve the energy conversion efficiency of the butterfly coil EMAT, the optimal parameters of the particular butterfly coil EMAT design modeled are $w_{2}$ = 25 mm, $h_{2}$ = 20 mm, $w_{1}$ = 0.2 mm, $h_{1}$ = 0.15 mm, n = 5, r = 2 mm, and $d_{1}$ = 0.1 mm. This combination of parameters is not included in the orthogonal array. In order to better observe the parameter changes, Table 5 shows the comparison of the parameters of the butterfly coil EMAT before and after the orthogonal test design. The simulation of the EMAT with these optimal parameters was performed, and the signal amplitude of the butterfly coil EMAT was calculated, as shown in Figure 7. The signal amplitude optimized by orthogonal test theory is 1.13 times greater than that before optimization.

## 5. The Modified Butterfly Coil EMAT

### 5.1. Configuration of the Modified EMAT

In this work, a modified EMAT is presented, as shown in Figure 8a, based on the optimized structural parameters of the butterfly coil EMAT. The traditional magnet combination consists of a permanent magnet, whereas the unique feature of the modified EMAT is that its magnet combination consists of three−square permanent magnets whose poles are opposite to each other. In addition, the overall dimensions of the two permanent magnets located on the outside are the same. It is noteworthy that the magnet volume and coil size of the modified EMAT remain equal to those of the EMAT optimized by orthogonal test theory. In Figure 8, the width of the middle magnet, the width of the two magnets located on the outside, and the width of the middle part of the butterfly coil are defined as $D_{1}$,$\;D_{2}$, and $L_{1}$, respectively. As the effective range of a magnet is controlled by the distance between the north and south poles, the static magnetic field of a conventional bulk magnet with north and south poles on opposite ends stretches out very widely. For the modified butterfly coil EMAT that consists of TSPM−OP, the north and south poles are on the same faces, creating a short distance between the poles and essentially keeping the static magnetic field from stretching [14]. The static magnetic field strength is increased, thus improving the low energy conversion efficiency of EMATs. In the manufacturing process of the modified EMAT, the permanent magnet is fixed in a mold with a specific magnetization orientation, as shown in Figure 8a. The butterfly coil is clamped and fixed to the magnet mold with coil stoppers and finally fixed to the sensor housing.

Since the middle part of the butterfly coil has the highest energy [31], it is necessary to focus on the influence of the ratio of the width of the middle magnet $D_{1}$ to the width of the middle part of the coil $L_{1}$ on the energy conversion efficiency of the EMAT. Consequently, we define ρ as the ratio of $D_{1}$ to $L_{1}$, that is:(9)$$\rho =\frac{D_{1}}{L_{1}}$$

Therefore, a model with different values was established. The ratio ρ of the width of the middle magnet to the width of the middle part of the coil ranged from 0.6 to 1.8 with a step of 0.1. The specific modeling parameters are shown in Table 6. It is worth noting that the width of the magnets on the left and right sides $D_{2}$ should be kept equal. The normalized amplitude for different values of ρ is calculated as shown in Figure 9. Surprisingly, unlike the Rayleigh wave EMAT, where the magnet is slightly narrower than the coil [36], the normalized amplitude increases linearly with an increasing ratio in the range of 0.6–1.2, but when this range is exceeded, the ratio and normalized amplitude are negatively exponential. It can be seen that, for the particular EMAT with aluminum as the tested specimen, a critical ρ of 1.2 is reached and the signal amplitude cannot increase further, but decreases. Therefore, the size of the magnets of the modified butterfly coil EMAT with TSPM−OP is determined to be $D_{1}$ = 6.72 mm and $D_{2}$ = 9.14 mm.

### 5.2. Analysis of Static Magnetic Field

Figure 10 shows the distribution of static magnetic flux density under the EMAT optimized by orthogonal test theory and the improved EMAT with TSPM−OP, and the white arrow indicates the direction of the static magnetic field. It can be seen in Figure 10a that the static magnetic flux density for the optimized butterfly coil EMAT by orthogonal test theory is mainly perpendicular to the surface of the magnet, with the maximum flux density at the edge of the magnet. Figure 10b shows that the improved EMAT with TSPM−OP has both horizontal and vertical fluxes, with the static magnetic flux density mainly concentrated inside the magnets. It can be seen that the static magnetic flux density of the optimized EMAT and the improved EMAT are 1.79 T and 3.4 T, respectively.

Figure 11 shows the distribution of static magnetic flux density on the surface of aluminum plates for the EMAT optimized by orthogonal test theory and the improved butterfly coil EMAT, *B_SX_* and *B_SY_* denote the X− and Y−components of the static magnetic flux density. In Figure 11a, *B_SX_* is zero at the center and reaches two peaks on both sides of the magnet, whereas *B_SY_* has a flat peak under the magnet face. Consequently, the static magnetic flux density of an optimized butterfly coil EMAT by orthogonal test theory is slightly larger towards the edges of the magnet. The maximum values of *B_SX_* and *B_SY_* are 0.663 T and 0.135 T, respectively. For the static magnetic flux density in the modified butterfly coil EMAT with TSPM−OP, in Figure 11b, *B_SX_* is zero at the center of the magnets and reaches a peak at the gap of the permanent magnet pair. The maximum values of *B_SX_* and *B_SY_* are 0.779 T and 0.52 T, respectively. Based on the above analysis, it is noteworthy that the static magnetic flux density component of the improved EAMT with TSPM−OP is much larger than that of the optimized butterfly coil EMAT, whether in the component parallel to the surface of the aluminum plate or perpendicular to the surface of the aluminum plate.

### 5.3. Analysis of Lorentz Force

The Lorentz forces generated by the optimized EMAT and the improved butterfly coil EMAT with TSPM−OP on the surface of the aluminum plate at *t* = 13 μs are shown in Figure 12a,b, respectively, and the direction of the red arrow indicates the direction of the Lorentz force. It can be seen that the maximum Lorentz force occurs below the coil, and the Lorentz force of the improved butterfly coil EMAT with TSPM−OP is increased by about 25%. It is worth noting that the direction of the Lorentz force differs between the improved butterfly coil EMAT with TSPM−OP and the butterfly coil EMAT optimized by orthogonal test theory due to the different directions of the static magnetic field being applied at the coil position.

### 5.4. Analysis of Signal Amplitude

To analyze the signal amplitudes of the optimized EMAT and the improved EMAT, their signal amplitudes were obtained, as shown in Figure 13. It can be seen that the signal amplitudes of the optimized EMAT and the improved EMAT are $1.1277\times 10^{-7}$ and $3.1567\times 10^{-7}$mm, respectively. The signal amplitude of the improved EMAT is increased by 1.8 times compared to the optimized EMAT. The normalized signal amplitude of the traditional EMAT, the EMAT optimized by orthogonal test theory, and the modified EMAT with TSPM−OP are shown in Figure 14. The signal amplitude of the modified EMAT with TSPM−OP is increased by 4.97 times compared to the traditional EMAT. This means that the modified EMAT has a higher energy conversion efficiency than the traditional EMAT when using the dimensions of the EMAT configuration.

Hence, it can be seen that the improved EMAT with TSPM−OP has significantly improved in terms of the static magnetic flux density, the Lorentz force, and the signal amplitude; that is, the improved butterfly coil EMAT with TSPM−OP is more conducive to improving its energy conversion efficiency.

## 6. Conclusions

The purpose of this work was to improve the energy conversion efficiency of the butterfly coil EMAT. The structural parameters of the EMAT were optimized and designed using orthogonal test theory. Based on this, a modified EMAT with TSPM−OP was proposed, and the influences on the energy conversion efficiency of the EMAT optimized by orthogonal test theory and the modified butterfly coil EMAT with TSPM−OP were compared and analyzed. The following conclusions are drawn:

(1)From the point of view of improving the energy conversion efficiency of the butterfly coil EMAT, $h_{2}$ and n have the greatest influence on the signal amplitude, followed by $w_{2}$, $h_{1}$, $w_{1}$, and $d_{1}$, whereas r has the least influence on the signal amplitude within the level of the selected factors. It can be seen that a decrease in $w_{2}$ and $d_{1}$, and an increase in $h_{2}$ and r, have a positive effect on the signal amplitude. In addition, when $w_{1}$ = 0.2, $h_{1}$ = 0.15, and n = 5, the amplitude of the ultrasonic signal is the largest. The design of the butterfly coil EMAT was optimized according to the above guidelines. The result indicates that, after the orthogonal test design for optimization, the signal amplitude of the EMAT was increased by 1.13 times.(2)A modified butterfly coil EMAT with TSPM−OP was proposed, and the signal amplitude tended to be maximized when the ratio ρ of the middle magnet width to the coil middle part width was approximately 1.2, beyond which the amplitude could not be increased. The modified butterfly coil EMAT with TSPM−OP results in a significant increase in the static magnetic flux density, the Lorentz force, and the signal amplitude. Compared to the traditional butterfly coil EMAT, the signal amplitude of the modified butterfly coil EMAT with TSPM−OP was increased by a factor of 4.97, thereby improving the energy conversion efficiency of the butterfly coil EMAT.

## Figures and Tables

**Figure 1 sensors-22-04985-f001:**
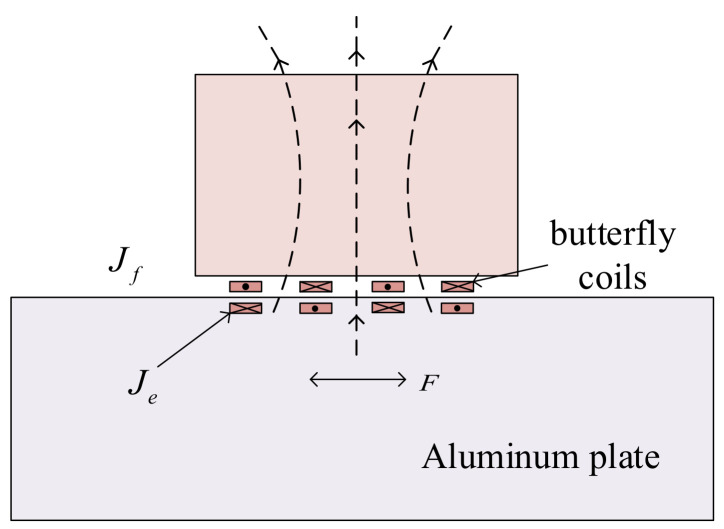
Working principal diagram of EAMTs.

**Figure 2 sensors-22-04985-f002:**
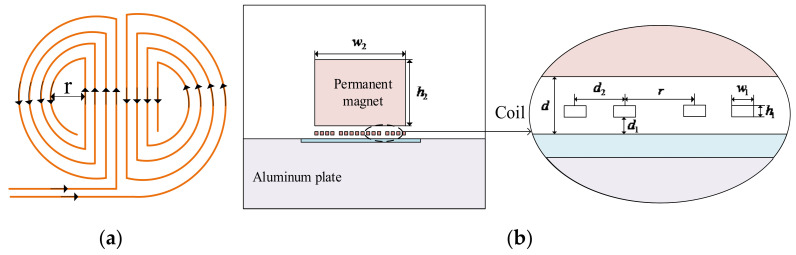
Schematic diagram of the butterfly coil EMAT: (**a**) top view of the butterfly coil; (**b**) 2−D model.

**Figure 3 sensors-22-04985-f003:**
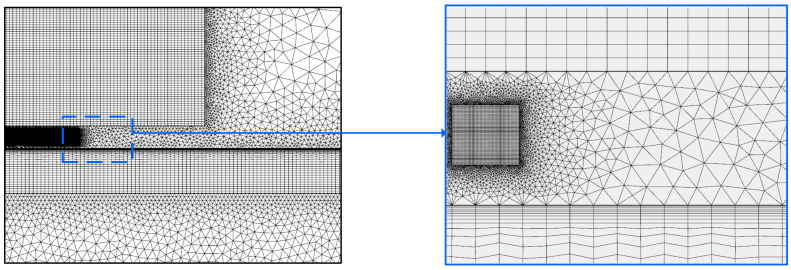
The schematic of the 2−D model in COMSOL.

**Figure 4 sensors-22-04985-f004:**
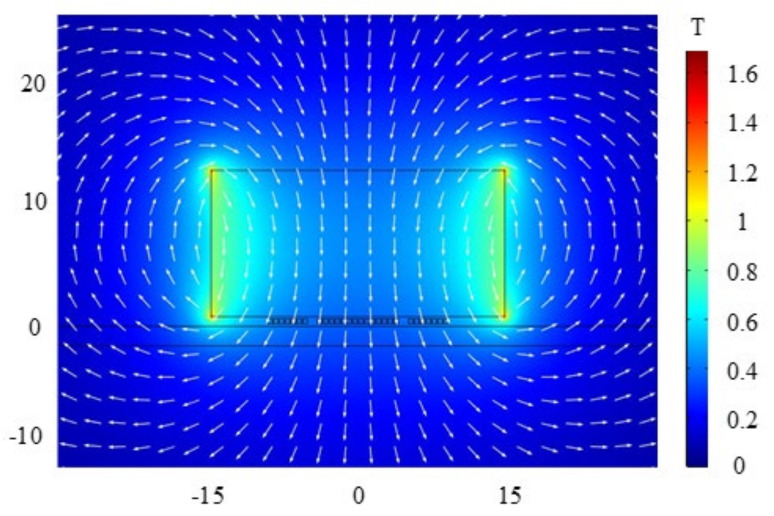
Static magnetic field distribution of the butterfly coil EMAT.

**Figure 5 sensors-22-04985-f005:**
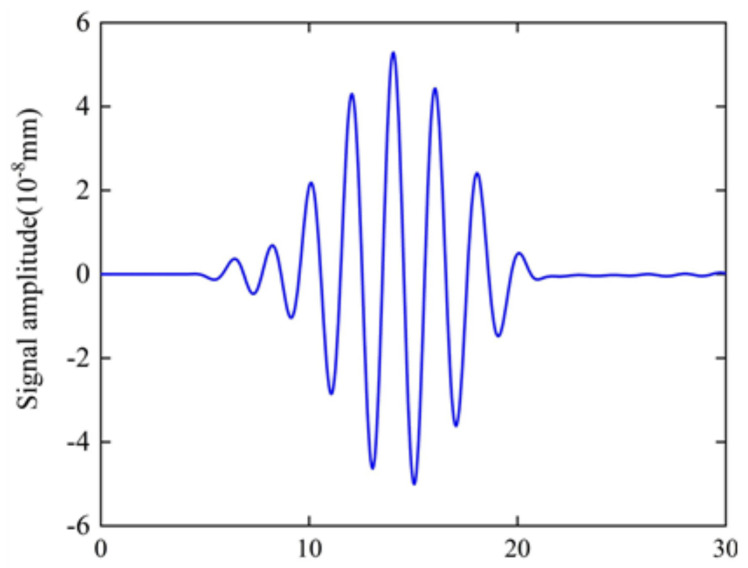
Signal amplitude at point P of the butterfly coil EMAT.

**Figure 6 sensors-22-04985-f006:**
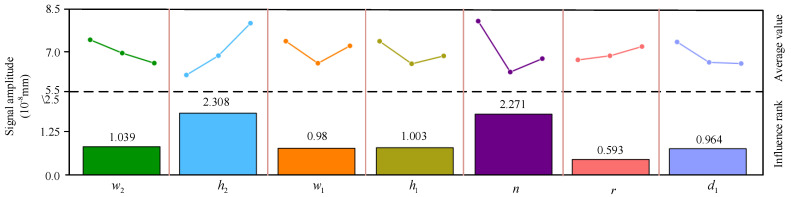
Range analysis of the results for the orthogonal test design; the column height represents the average value of each level and the influence rank for each factor.

**Figure 7 sensors-22-04985-f007:**
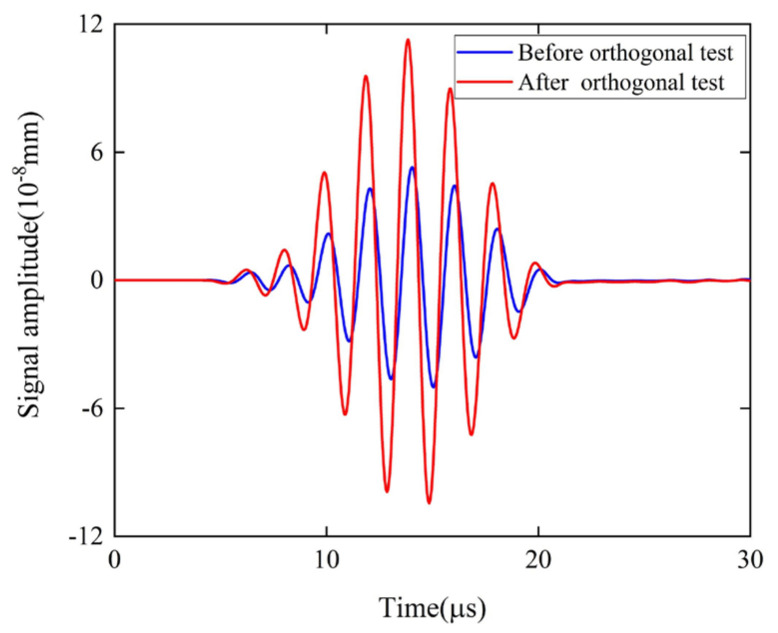
Comparison of the signal amplitude of the butterfly coil EMAT before and after the orthogonal test design.

**Figure 8 sensors-22-04985-f008:**
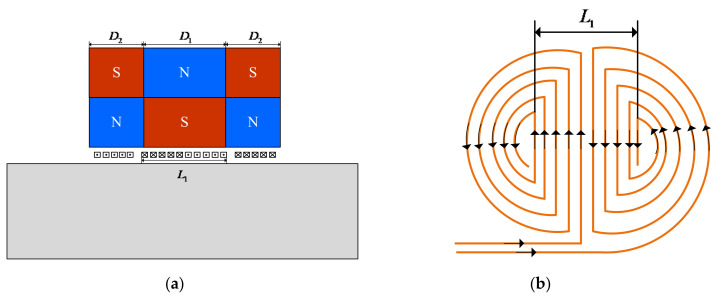
Geometrical parameters of the modified butterfly coil EMAT: (**a**) 2−D model; (**b**) top view of the butterfly coil.

**Figure 9 sensors-22-04985-f009:**
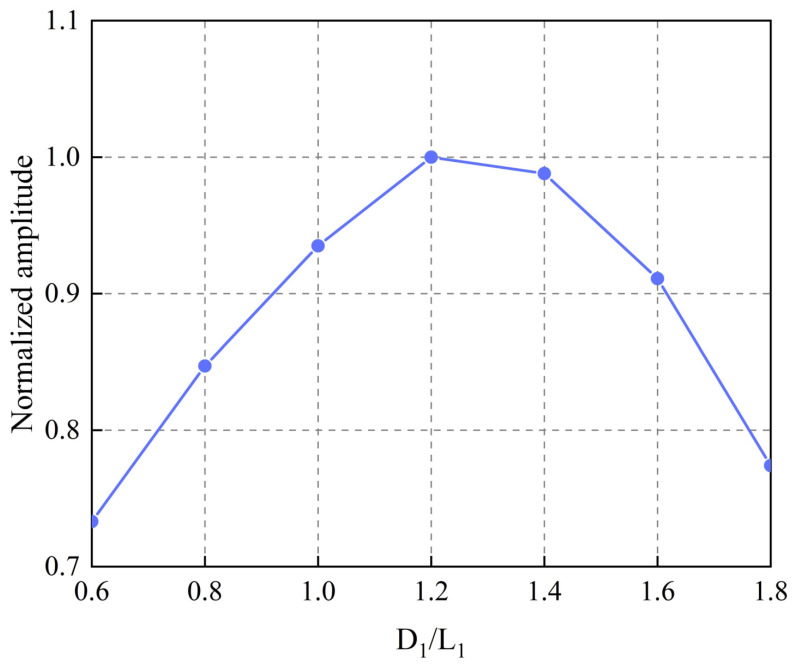
Normalized amplitude for the butterfly coil EMAT modeling with various values of ρ.

**Figure 10 sensors-22-04985-f010:**
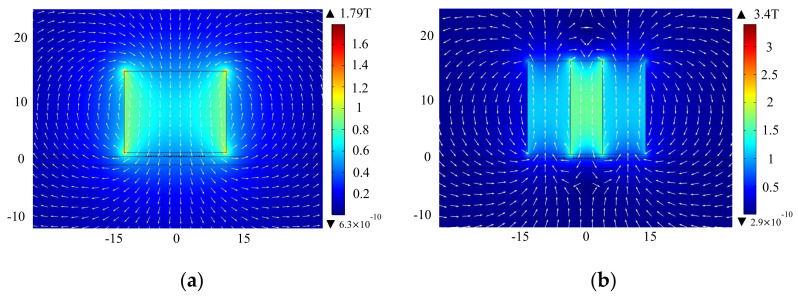
The distribution of static magnetic flux density from (**a**) the EMAT optimized by orthogonal test theory and (**b**) the improved butterfly coil EMAT.

**Figure 11 sensors-22-04985-f011:**
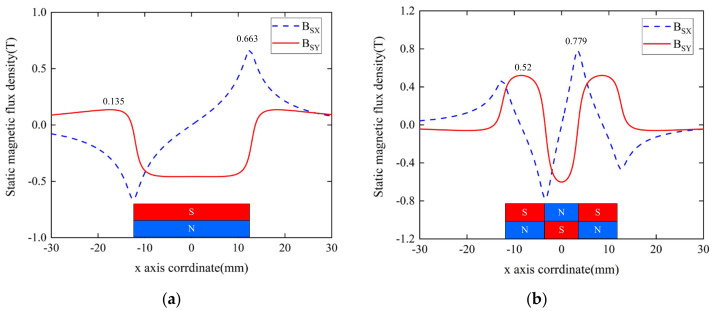
The distribution of static magnetic flux density on the surface of aluminum plates from (**a**) the EMAT optimized by orthogonal test theory and (**b**) the improved butterfly coil EMAT.

**Figure 12 sensors-22-04985-f012:**
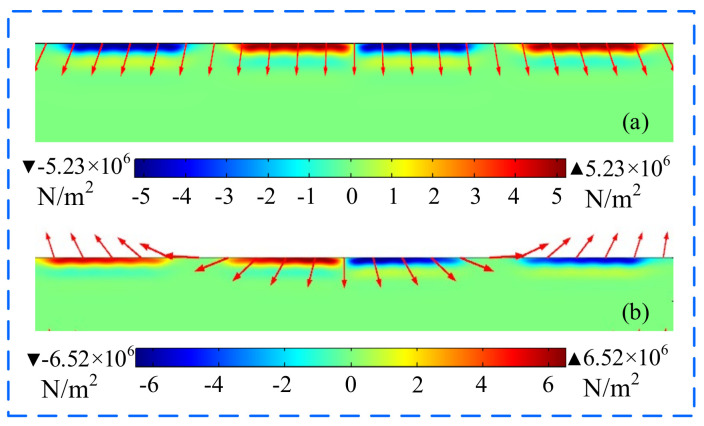
The distribution of Lorentz forces in the aluminum plate from (**a**) the EMAT optimized by orthogonal test theory and (**b**) the improved butterfly coil EMAT.

**Figure 13 sensors-22-04985-f013:**
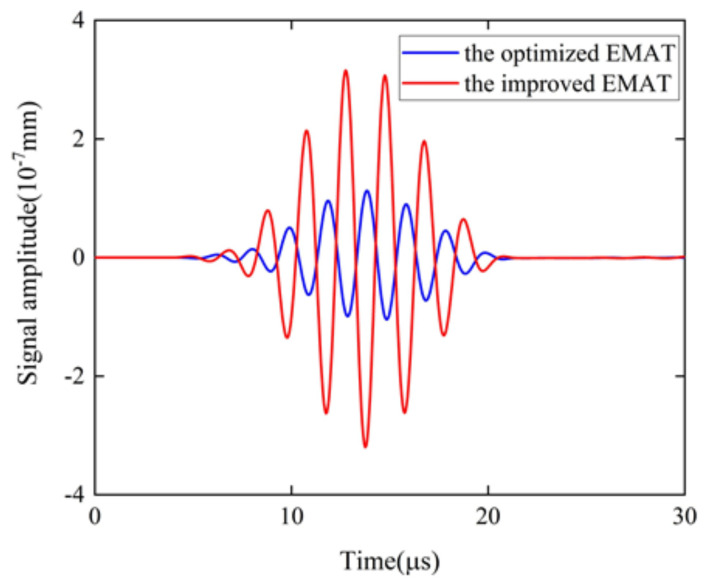
Comparison of the signal amplitude of the EMAT optimized by orthogonal test theory and the improved butterfly coil EMAT.

**Figure 14 sensors-22-04985-f014:**
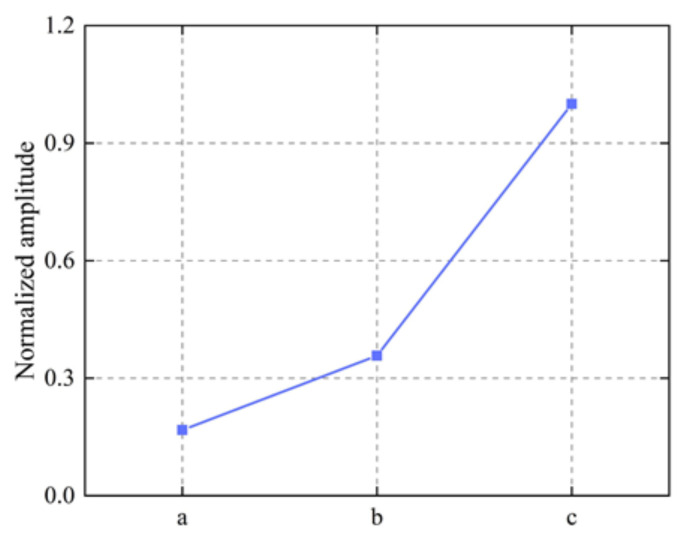
Comparison of the normalized amplitude: (**a**) the traditional EMAT; (**b**) the EMAT optimized by orthogonal test theory; (**c**) the modified EMAT with TSPM−OP.

**Table 1 sensors-22-04985-t001:** Parameters of the 2−D FEM used in this work.

Object	Parameters	Symbol	Value
Permanent magnet	Width	$w_{2}$	30 mm
	Height	$h_{2}$	15 mm
	Magnetic flux density	$B_{s}$	1.4 T
	Lift−off distance	d	1 mm
coil	Width	$w_{1}$	0.5 mm
	Height	$h_{1}$	0.45 mm
	Inner diameter	r	1.5 mm
	Number of turns	n	7
	Lift−off distance Interval	$d_{1}$ $d_{2}$	0.3 mm 0.6 mm
Al	Width	W	100 mm
	Thickness	H	50 mm
	Mass density	ρ	2700 kg/m^3^
	Electrical conductivity	σ	3.77 × 10^7^ S/m
	Young’s modulus	E	70 × 10^9^ Pa
	Passion’s ratio	μ	0.33
Excitation Current	Current Frequency	I f	100 A 500 kHz

**Table 2 sensors-22-04985-t002:** Ranges of variations in EMAT parameters.

Levels	$w_{2}\;(\mathrm{mm})$	$h_{2}\;(\mathrm{mm})$	$w_{1}\;(\mathrm{mm})$	$h_{1}\;(\mathrm{mm})$	n	r (mm)	$d_{1}\;(\mathrm{mm})$
1	25	10	0.2	0.15	5	1	0.1
2	30	15	0.35	0.3	7	1.5	0.2
3	35	20	0.5	0.45	9	2	0.3

**Table 3 sensors-22-04985-t003:** L_18_ (3^7^) orthogonal array testing for butterfly coil EMATs.

Run	$w_{2}\;(\mathrm{mm})$	$h_{2}\;(\mathrm{mm})$	$w_{1}\;(\mathrm{mm})$	$h_{1}\;(\mathrm{mm})$	n	r (mm)	$d_{1}\;(\mathrm{mm})$	Signal Amplitude (10^−8^ mm)
1	25	10	0.2	0.15	5	1	0.1	8.537
2	25	15	0.35	0.3	7	1.5	0.2	4.417
3	25	20	0.5	0.45	9	2	0.3	7.937
4	30	10	0.2	0.3	7	2	0.3	5.002
5	30	15	0.35	0.45	9	1	0.1	6.203
6	30	20	0.5	0.15	5	1.5	0.2	10.02
7	35	10	0.35	0.15	9	1.5	0.3	4.731
8	35	15	0.5	0.3	5	2	0.1	8.414
9	35	20	0.2	0.45	7	1	0.2	6.639
10	25	10	0.5	0.45	7	1.5	0.1	6.793
11	25	15	0.2	0.15	9	2	0.2	8.614
12	25	20	0.35	0.3	5	1	0.3	9.104
13	30	10	0.35	0.45	5	2	0.2	6.492
14	30	15	0.5	0.15	7	1	0.3	5.864
15	30	20	0.2	0.3	9	1.5	0.1	8.256
16	35	10	0.5	0.3	9	1	0.2	4.237
17	35	15	0.2	0.45	5	1.5	0.3	7.463
18	35	20	0.35	0.15	7	2	0.1	7.684

**Table 4 sensors-22-04985-t004:** Analysis of the results of the orthogonal test design.

Amplitude		$w_{2}\;(\mathrm{mm})$	$h_{2}\;(\mathrm{mm})$	$w_{1}\;(\mathrm{mm})$	$h_{1}\;(\mathrm{mm})$	n	r(mm)	$d_{1}\;(\mathrm{mm})$
Signal amplitude (10^−8^ mm)	$k_{1}$	7.567	5.965	7.419	7.575	8.338	6.764	7.648
	$k_{2}$	6.973	6.829	6.439	6.572	6.067	6.947	6.737
	$k_{3}$	6.528	8.273	7.211	6.921	6.663	7.357	6.684
	R	1.039	2.308	0.98	1.003	2.271	0.593	0.964
Influence rank	$h_{2}$ (2.308) > n (2.271) > $w_{2}$ (1.039) > $h_{1}$ (1.003) > $w_{1}$ (0.98) > $d_{1}$ (0.964) > r (0.593)							

**Table 5 sensors-22-04985-t005:** Comparison of the parameters of the butterfly coil EMAT before and after the orthogonal test design.

EMAT	$w_{2}\;(\mathrm{mm})$	$h_{2}\;(\mathrm{mm})$	$w_{1}\;(\mathrm{mm})$	$h_{1}\;(\mathrm{mm})$	n	r (mm)	$d_{1}\;(\mathrm{mm})$
The traditional EMAT	30	15	0.5	0.45	7	1.5	0.3
The optimized EMAT	25	20	0.2	0.15	5	2	0.1

**Table 6 sensors-22-04985-t006:** Magnet size at different ratios.

Level	ρ=0.6	ρ=0.8	ρ=1.0	ρ=1.2	ρ=1.4	ρ=1.6	ρ=1.8
$D_{1}$	3.36 mm	4.48 mm	5.6 mm	6.72 mm	7.84 mm	8.96 mm	10.08 mm
$D_{2}$	10.82 mm	10.26 mm	9.7 mm	9.14 mm	8.58 mm	8.02 mm	7.46 mm
$L_{1}$	5.6 mm	5.6 mm	5.6 mm	5.6 mm	5.6 mm	5.6 mm	5.6 mm

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
