# Peer review of "Numerical Study and Optimal Design of the Butterfly Coil EMAT for Signal Amplitude Enhancement"

_sensors, 2022, doi:10.3390/s22134985_

Round 1
Reviewer 1 Report
The authors present a modified butterfly coil design for EMAT. The results are interesting and are of interest to the community. I recommend minor corrections to the authors:
1. Introduction: please introduce what is orthogonal test theory and existing literature in this area. The need for orthogonal test theory is also missing. i recommend clearly stating why and how this will help.
2. Fig. 7, maybe insert a table that shows the two parameters; the optimized parameters and the non-optimized parameters.
3. Comment on the actual manufacturability of these designs. can the modified EMAT be realized using current manufacturing methodologies?
4. i recommend adding the term: "numerical study" in your title. The current title is misleading.
5. There are small grammatical and sentence formation errors. I recommend a close grammar-edit of the article.
Reviewer 2 Report
This paper describes a topic on the low energy conversion efficiency of electromagnetic acoustic transducers (EMATs) is a critical issue in nondestructive testing applications. To overcome this shortcoming, a butterfly coil EMAT is developed and optimized by numerical simulation based on a 2-D finite element model. Firstly, the effect of the structural parameters of the butterfly coil EMAT is investigated by orthogonal test theory. Then, a modified butterfly coil EMAT is presented, which consists of three- square permanent magnets with opposite polarity (TSPM-OP), for enhancing the signal amplitude. Finally, the signal amplitude obtained from the three types of EMATs, that is, the traditional one, the optimized one by orthogonal test theory, and the modified one with TSPM-OP, are analyzed and compared. The results show that the signal amplitude launched by the modified butterfly coil EMAT with TSPM-OP can be increased by 4.97 times compared to the traditional butterfly coil EMAT. Some key aspects are left out. Therefore, the paper needs mandatory revise before it can be accepted. Meanwhile, the following comments should be addressed before publications.
1. It is strongly recommended that the authors should mention clearly the newly developed and /or found point of in section introduction, compared with papers already reported in this field.
2. The authors present the results of nondestructive testing (NDT) methods are currently widely used in industry and structural health monitoring, but the technical and academic descriptions are still deficient. The authors should provide more technical and academic descriptions on what different/ effect compared with existed (NDT) works.
3. The authors should compare clearly what the difference for the magnetic sensor is installed, how to improve the energy conversion efficiency of the butterfly coil EMAT, the optimal parameters of the particular butterfly coil EMAT design, and how to define and change the sensitivity?
4. The authors should discuss what reliability mechanisms on particular butterfly coil EMAT sensing design.
5. How to identify that forming a higher sensitivity at t particular butterfly coil EMAT sensing design?
6. The authors should show and compare the quantitative data of sensitivity.
Adding the references from 2021 to 2022 is recommended.
